# RapidDock: Unlocking Proteome-scale Molecular Docking

## Abstract

Accelerating *molecular docking* – the process of predicting how molecules bind to protein targets – could boost small-molecule drug discovery and revolutionize medicine. Unfortunately, current molecular docking tools are too slow to screen potential drugs against all relevant proteins, which often results in missed drug candidates or unexpected side effects occurring in clinical trials. To address this gap, we introduce RapidDock, an efficient transformer-based model for blind molecular docking. RapidDock achieves at least a $100\times$ speed advantage over existing methods without compromising accuracy. On the Posebusters and DockGen benchmarks, our method achieves $52.1\%$ and $44.0\%$ success rates (RMSD $<2$Å), respectively. The average inference time is $0.04$ seconds on a single GPU, highlighting RapidDock's potential for large-scale docking studies. We examine the key features of RapidDock that enable leveraging the transformer architecture for molecular docking, including the use of relative distance embeddings of 3D structures in attention matrices, pre-training on protein folding, and a custom loss function invariant to molecular symmetries. We make the model code and weights publicly available.

## 1 Introduction

Accelerating the drug discovery process could revolutionize medicine. As most novel drugs are small molecules (Kinch et al., 2024), various deep learning methods have been proposed to streamline the process of docking such molecules to druggable protein targets (Abramson et al., 2024; Corso et al., 2024; 2022; Qiao et al., 2024). While impressive, none of these methods is both *accurate* and *fast*.

In fact, to obtain a comprehensive view of its effects, a molecule needs to be screened against thousands of proteins (Sjöstedt et al., 2020). However, state-of-the-art methods (Corso et al., 2024; 2022; Abramson et al., 2024; Qiao et al., 2024) report run-times on the scale of seconds per protein on a single GPU. Consequently, screening a relatively small database of one million molecules (Yu et al., 2023) against the *proteome*, i.e., all proteins in the human body, would take years, even with hundreds of GPUs. Such time-frames are unacceptable in the drug development process.

To address this challenge, we introduce RapidDock, a transformer-based model that performs molecular docking in a single forward pass, in hundredths of a second on a single GPU. RapidDock performs blind docking, using unbound, possibly computationally folded proteins, so it can be applied to unexplored protein targets. Given the 3D structure of a protein and a molecule, our method predicts all pairwise distances in the resulting protein-molecule complex, including the molecule's atom-atom distances and atom-amino acid distances.

RapidDock achieves success rates, i.e., RMSD $< 2$Å, of $52.1\%$ on the Posebusters benchmark (Buttenschoen et al., 2024) and $44.0\%$ on the DockGen benchmark (Corso et al., 2024), with 0.03 and 0.05 seconds average runtimes, respectively. Our method significantly outperforms the recent DiffDock-L (Corso et al., 2024) (22.6% success rate, 35.4 seconds on DockGen) or the NeuralPLexer (Qiao et al., 2024) (27.4% success rate, 3.77 seconds on Posebusters). Because of its accuracy and speed, RapidDock can enable novel use cases and research directions. For example, with RapidDock, docking ten million molecules to all human proteins on a cluster with 512 GPUs would take nine days, compared to about 20 years with DiffDock-L or even 200 years required with a computationally-intensive method like AlphaFold-3 (Abramson et al., 2024).

We supplement the analysis of RAPIDDOCK's performance with a thorough study of design choices, which include: operating on relative distances, pre-training on a protein folding task, and a special treatment of multiple molecular conformations in the loss function.

In summary, our contributions are as follows:

- We introduce RAPIDDOCK, a first-in-class transformer-based model capable of accurate high-throughput molecular docking. We open the model for public use at `github_and_huggingace_links_will_be_avaiable_in_the_final_version`.

- We run RAPIDDOCK on the most challenging benchmarks. The model docks 52.1% of examples with RMSD < 2Å on Posebusters, and 44.0% on DockGen, which places it among the most accurate methods. Importantly, inference on a single GPU takes only 0.04 seconds on average, making RAPIDDOCK at least 100× faster than comparable methods.

- We provide ablations and describe the design choices that led to RAPIDDOCK's success.

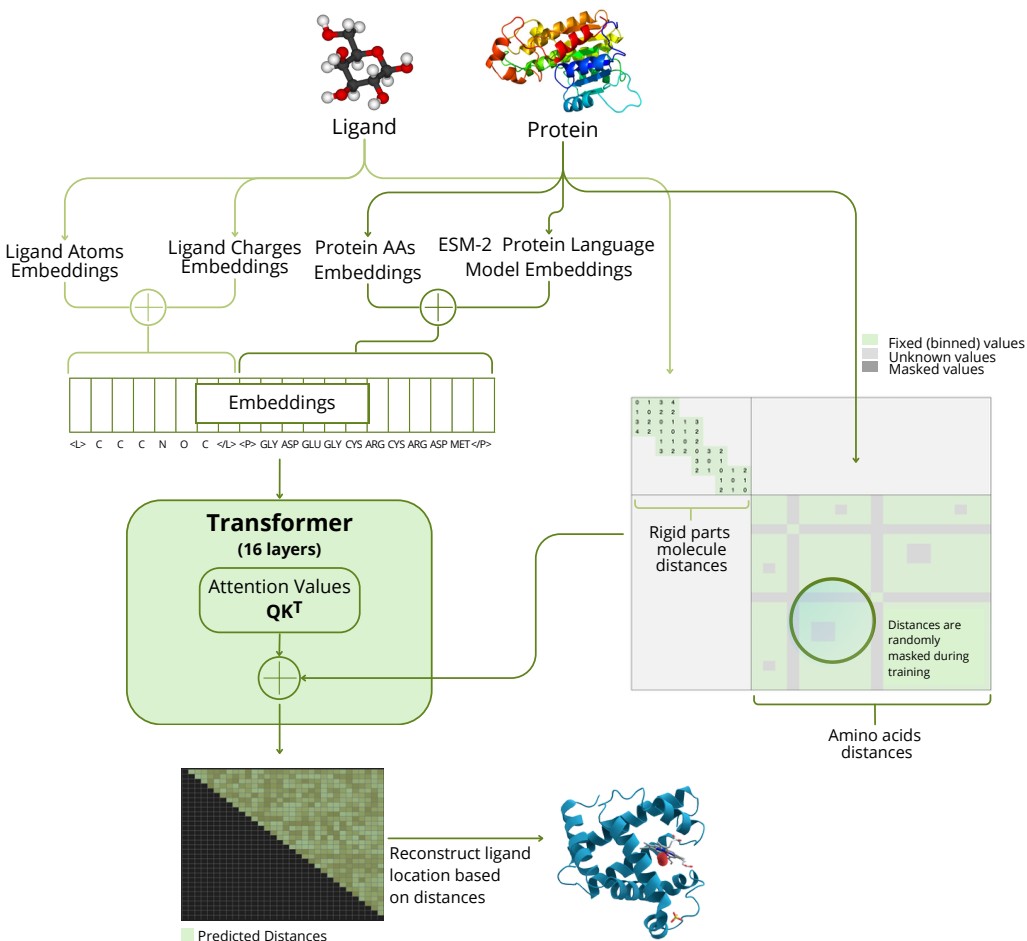

Figure 1: RAPIDDOCK architecture overview. The molecule is represented by a sequence of its atoms and the corresponding matrix of distances. The protein is represented by its amino acid sequence and its matrix of distances. Learnable embeddings of these distance matrices are added to the attention matrices. Additionally, the model uses ESM-2 embeddings (Lin et al., 2023) to improve its protein representation and embeddings of atom charges to improve its molecule representation.

## 2 METHOD

RAPIDDOCK, illustrated in Figure 1, is based on the transformer encoder architecture (Vaswani, 2017) and returns the positions of atoms in the docked molecule. RAPIDDOCK represents the protein and molecule spatial structures by distance matrices whose embeddings are added to the attention matrices. Because RAPIDDOCK operates on distances only, it is invariant to translations, reflections, or rotations, which we found crucial for accurate docking. Below we present the details.

### 2.1 MOLECULE AND PROTEIN REPRESENTATION

The input to RAPIDDOCK consists of a sequence of tokens corresponding to molecule atoms and protein amino acids. The 3D structures of the protein and molecule are represented by learnable embeddings of their respective distance matrices. Additionally, we include the information on the molecule's atom charges because of the need to model the electrostatic interactions governing the docking process. We also utilize amino acid embeddings obtained from the protein language model ESM-2 (Lin et al., 2023).

**Atom and amino acid dictionary** We have 26 amino acid tokens (20 standard amino acids and 6 tokens for non-standard or unknown residues) and 56 atom tokens (all atoms likely to be found in drug molecules, i.e., elements lighter than uranium, excluding noble gases or reactive alkali metals).

**Ligand atom charges** To account for electrostatic interactions, we include information about atom charges in the molecules. For each atom we include its partial charge (Gasteiger charges), which is the distribution of the molecule electron density spread over the ligand atoms. The values of charges are embedded (see below) and added to the atom embeddings.

**Protein language embeddings** To improve the protein representation, we use embeddings from the 650M ESM-2 model Lin et al. (2023), one embedding per amino acid. These embeddings are added to the amino acid embeddings (to match the dimensionality, we use a linear projection layer).

**Molecule rigid distance matrix** The ligand spatial structure is represented by the distance matrix $\boldsymbol{D}^l$ between its heavy atom coordinates $\boldsymbol{x}^l \in \mathbb{R}^{L \times 3}$, where $L$ is the number of heavy atoms in the ligand. Only the fixed distances across the molecule's possible conformations are recorded, and others are denoted by a special value of $-1$, which indicates that the corresponding attention matrix entries should not be modified, see below. The moleule's geometric representation thus distinguishes the rigid and moving parts of the molecule, which is important for modeling its allowable movements. To obtain the matrix, we generate 96 molecule conformations in total, using two methods: 64 from RDKit's built-in algorithm based on MMFF optimization and 32 from the torsion-based algorithm described in (Zhou et al., 2023). The distances are averaged across all conformations and only elements with a standard deviation below 0.3 Å are recorded.

**Protein distance matrix** To represent the protein's 3D structure, we extract the positions of alpha-carbon atom for each amino acid $\boldsymbol{x}^p \in \mathbb{R}^{P \times 3}$, where $P$ is the number of amino acids in the protein, and calculate the distance matrix $\boldsymbol{D}^p = \left( \|\boldsymbol{x}_i^p - \boldsymbol{x}_j^p\| \right)_{1 \leq i,j \leq P}$.

### 2.2 MODEL ARCHITECTURE

RAPIDDOCK model is based on standard transformer (in our implementation, we use Mistral (Jiang et al., 2023) architecture), with full attention mask used instead of the causal one due to the non-autoregressive nature of the docking task. Below, we describe the modifications to the model: learnable attention biases based on the distance matrices, attention scalers, and learnable charge embeddings.

**Distance biases matrices** Given the input distance matrices $\boldsymbol{D}^l$ for ligand and $\boldsymbol{D}^p$ for protein, we construct a block matrix $\boldsymbol{D} = \begin{bmatrix} \boldsymbol{D}^l & -1 \\ -1 & \boldsymbol{D}^p \end{bmatrix}$. Further, following (Raffel et al., 2020), we discretize

the distances into 257 buckets with:

$$b(x) := \begin{cases} \lfloor 8x \rfloor + 1 & \text{if } 0 \le x \le 16, \\ \min\left(256, \left\lfloor 128 + 128\frac{\ln\left(\frac{8x}{128}\right)}{\ln(6)} \right\rfloor\right) + 1 & \text{if } x > 16, \\ 257 & \text{if } x = -1. \end{cases}$$

Intuitively, the discretization is finer for small distances and coarser for larger ones, for which high accuracy is not necessary. The chosen resolution should be sufficient to capture the nuanced properties of the molecule's geometry, e.g., to distinguish the double carbon bond from the single carbon bond. The maximum representable distance, on the other hand, should allow capturing long-range interactions within the protein. The discretized values are used to construct distance bias matrices. For each attention head $l$ we put

$$\boldsymbol{B}^l = \left(\boldsymbol{E}_{b(\boldsymbol{D}_{i,j}),l}\right)_{1 \le i,j \le L+P},$$

where $\boldsymbol{E} \in \mathbb{R}^{257 \times H}$ is a learnable embedding matrix and $H$ is the number of attention heads.

**RAPIDDOCK attention**   As the distance matrices describe a pairwise property of the tokens (atoms or amino acids), we chose to embed them within the attention matrices. We modify the attention mechanism in two ways (marked in blue in (1)). First, we multiply the attention scores corresponding to input pairs with known distances (i.e. ligand-ligand within a rigid part and protein-protein) by a learnable scalar $s_m$, one for each layer $m$. Second, we inject distance information by adding $z_m \cdot \boldsymbol{B}^l$, where $z_m$ is another learnable scalar for each layer $m$ and $\boldsymbol{B}^l$ is the distance bias matrix. The scalars $s\_m$ and $z\_m$ control to what extent the final score is affected by the distance matrix bias. The final attention formula for the $l$-th head in the $m$-th layer is defined as follows:

$$\text{Attention}(\boldsymbol{Q}, \boldsymbol{K}, \boldsymbol{V}) = \text{softmax}\left(\frac{\boldsymbol{Q}\boldsymbol{K}^T}{\sqrt{d_k}} \odot \boldsymbol{S}_m + \boldsymbol{z}_m \cdot \boldsymbol{B}^l\right)\boldsymbol{V}, \tag{1}$$

where $\boldsymbol{Q}, \boldsymbol{K}, \boldsymbol{V}$ are the query, key, and value matrices, $d_k$ is the dimension of the key vectors and

$$\boldsymbol{S}_m = \left(\begin{cases} 1 & \text{if } \boldsymbol{D}_{i,j} = -1 \text{ (unknown distance)} \\ s_m & \text{otherwise} \end{cases}\right)_{1 \le i,j \le L+P.}$$

**Charge embeddings**   We discretize charge values into 22 buckets using the following formula

$$c(x) = \lfloor 10 \cdot \max(\min(x,1), -1) + 11.5 \rfloor,$$

with the $0$-th bucket reserved for unknown charges. Our model learns an embedding matrix $\boldsymbol{C} \in \mathbb{R}^{22 \times D}$, where $D$ is the dimension of the model's hidden states.

**Molecule and protein input**   The model input is the concatenation: $\boldsymbol{X} = \begin{bmatrix} \boldsymbol{X}^l; \boldsymbol{X}^p \end{bmatrix} \in \mathbb{R}^{(L+P) \times D}$. For the $i$-th molecule atom, $\boldsymbol{X}_i^l$ is the sum of the learnable atom embedding and the charge embedding. For the $j$-th amino acid, $\boldsymbol{X}_j^p$ is the sum of the amino acid embedding and the ESM-2 embedding.

### 2.3   TRAINING LOSS FUNCTION

The model outputs predicted distance matrices, $\hat{\boldsymbol{D}}^l$ for the ligand, $\hat{\boldsymbol{D}}^p$ for the protein and $\hat{\boldsymbol{D}}^{lp}$ for ligand-protein distances. More precisely, a linear layer is applied to the transformer outputs, resulting in protein and ligand representations $\hat{\boldsymbol{x}}^l \in \mathbb{R}^{L \times H}$ and $\hat{\boldsymbol{x}}^p \in \mathbb{R}^{P \times H}$, where $H$ is the output dimension, equal to 16 in our experiments. The distance matrices are then defined as $\hat{\boldsymbol{D}}^l = \left(\|\hat{\boldsymbol{x}}_i^l - \hat{\boldsymbol{x}}_j^l\|\right)_{1 \le i,j \le L}$, $\hat{\boldsymbol{D}}^p = \left(|\hat{\boldsymbol{x}}_i^p - \hat{\boldsymbol{x}}_j^p\|\right)_{1 \le i,j \le P}$, $\hat{\boldsymbol{D}}^{lp} = \left(\|\hat{\boldsymbol{x}}_i^l - \hat{\boldsymbol{x}}_j^p\|\right)_{1 \le i \le L, 1 \le j \le P}$. We apply the L1 loss to them during training. In order to keep the model focused on close interactions, we ignore the loss on terms that have both the predicted and ground-truth distances above 20Å. The total loss consists of three parts: ligand loss $\mathcal{L}^l$, protein loss $\mathcal{L}^p$ and docking loss $\mathcal{L}^d$:

$$\mathcal{L} = \mathcal{L}^l + \mathcal{L}^p + \mathcal{L}^d, \text{ where}$$

$$\mathcal{L}^l = \sum_{i<j} |\hat{\boldsymbol{D}}^l_{i,j} - \boldsymbol{D}^l_{i,j}| \cdot \mathbb{1}(\min(\hat{\boldsymbol{D}}^l_{i,j}, \boldsymbol{D}^l_{i,j}) < 20),$$

$$\mathcal{L}^p = \sum_{i<j} |\hat{\boldsymbol{D}}^p_{i,j} - \boldsymbol{D}^p_{i,j}| \cdot \mathbb{1}(\min(\hat{\boldsymbol{D}}^p_{i,j}, \boldsymbol{D}^p_{i,j}) < 20),$$

$$\mathcal{L}^d = \sum_{i,j} |\hat{\boldsymbol{D}}^{lp}_{i,j} - \boldsymbol{D}^{lp}_{i,j}| \cdot \mathbb{1}(\min(\hat{\boldsymbol{D}}^{lp}_{i,j}, \boldsymbol{D}^{lp}_{i,j}) < 20),$$

where $\boldsymbol{D}^{lp} = \left(\|\boldsymbol{x}^l_i - \boldsymbol{x}^p_j\|\right)_{1\leq i\leq L, 1\leq j\leq P}$ is the distance matrix between true atom and protein coordinates, whereas $\boldsymbol{D}^l$ and $\boldsymbol{D}^p$ are as defined above.

In case of molecules with permutation symmetries, such as benzene, a single docked pose might be represented by several sets of atom positions, and thus different distance matrices. In such a case, the loss function may not assign the smallest value to the correct conformation. To alleviate this issue, we apply the loss function to all of the possible permutations of the molecule and use only the best match (the one with the smallest loss value) in the backward pass. This prevents the model from having to guess the specific atom order used in the docked pose. This procedure is similar to the Permutation Loss introduced in (Zhu et al., 2022), with our approach applying permutations to the ground truth (labels) instead of the predictions.

### 2.4 LIGAND RECONSTRUCTION

The last transformer layer outputs the predicted distance matrices. To reconstruct the 3-dimensional coordinates of the ligand atoms $\tilde{\boldsymbol{x}}^l \in \mathbb{R}^{l\times 3}$, RAPIDDOCK uses L-BFGS (Liu & Nocedal, 1989) algorithm with the following objective function defined on the predicted ligand-ligand and ligand-protein distance matrices:

$$\mathcal{L}^r = \sum_{i,j} |\|\tilde{\boldsymbol{x}}_i - \boldsymbol{x}^p_j\| - \hat{\boldsymbol{D}}^{lp}_{i,j}| \cdot \mathbb{1}(\hat{\boldsymbol{D}}^{lp}_{i,j} < 20) + \sum_{i<j} |\|\tilde{\boldsymbol{x}}_i - \tilde{\boldsymbol{x}}_j\| - \hat{\boldsymbol{D}}^l_{i,j}| \cdot \mathbb{1}(\hat{\boldsymbol{D}}^l_{i,j} < 20), \quad (2)$$

with initial guess for molecule atoms defined as the weighted mean of the four closest amino acids based on the predicted distance matrix $\hat{\boldsymbol{D}}^{lp}$. The computational cost of this operation is comparable to the transformer forward pass and is included in our runtime results. For details, see Appendix A.5.

## 3 EXPERIMENTS SETUP

### 3.1 DATASETS

**Training and test datasets**  Following (Corso et al., 2024), RAPIDDOCK is trained on PDBBind (Liu et al., 2017) and BindingMOAD (Hu et al., 2005) and tested on Posebusters (Buttenschoen et al., 2024) and DockGen (Corso et al., 2024) datasets. In addition, we augment the training dataset by computationally generated apostructures – unbound protein structures in the absence of a ligand – for about 30% of the training examples using AlphaFold-2 server, so that the model can see both the unbound and ground-truth bound proteins (also called *holostructures*). The apostructures are only used for defining the embeddings of the input distance matrices, while the loss is always computed based on the holostructures. This way, we allow the model to learn to deform the protein during the docking process.

**Ligands preparation**  The 3D coordinates of ligands are read from their PDB files and their graph structure is matched with their CCD reference. Entries for which such matching fails are filtered out. Ligands with more than 128 heavy atoms are filtered out. Whenever calculating the ligand rigid distance matrix is impossible because of a failure in calculating conformations or force fields, such entry is removed.

### 3.2 TRAINING PROCEDURE

RAPIDDOCK's transformer architecture has 16 layers, each with 4 attention heads and a hidden size of 512, resulting in approximately 60 million parameters. We found that such "deep and narrow" architecture performed better than shallower-but-wider models with the same number of parameters.

Since similar physico-chemical principles govern both protein folding and ligand binding, we pre-trained the model on a protein folding task with the hope that it would improve the docking performance. We used approximately 440k structures generated by AlphaFold-2 on the SwissProt protein database (Varadi et al., 2024; Gasteiger et al., 2001). More precisely, the model was trained to predict distances between amino acids, with a masking factor of 97% applied to the input distances, which effectively simulates the protein folding task.

Following pre-training, the model was fine-tuned on the molecular docking task on our training dataset. In total, the training process took about three days on a machine with eight A100 GPUs. The details are described in A.4

## 4 EXPERIMENTS

We evaluate RAPIDDOCK on two recent challenging benchmarks that were not part of the training dataset: the DockGen benchmark (Corso et al., 2024), and the Posebusters dataset (Buttenschoen et al., 2024). These benchmarks cover a wide range of structures across various protein domains. We calculate the heavy-atom Root Mean Square Deviation (RMSD) between the predicted ligand and the crystal (ground-truth) ligand atoms. We report both the median RMSD and the percent of examples with RMSD $< 2$Å (denoted % RMSD $< 2$Å), the latter being a widely-used threshold for considering a molecule to be correctly docked (Bell & Zhang, 2019). A more detailed definition of RMSD is described in A.2.

### 4.1 MAIN RESULTS

**Accuracy and speed** We compare RAPIDDOCK, both in terms of accuracy and runtimes, to a classical docking tool SMINA (Koes et al., 2013) and three recent state-of-the-art blind docking methods: AlphaFold-3 (Abramson et al., 2024), NeuralPLexer (Qiao et al., 2024), and DiffDock-L (Corso et al., 2024), see Table 1. RAPIDDOCK is more accurate than DiffDock-L and about three orders of magnitude faster. While the speed advantage over NeuralPLexer is less pronounced (about $100\times$ faster), RAPIDDOCK has an even larger accuracy advantage over that model. Finally, RAPIDDOCK is a staggering four orders of magnitude faster than AlphaFold-3. Although AlphaFold-3 achieves the highest accuracy, it is computationally intensive and therefore less practical for high-throughput screening studies with thousands of compounds.

Overall, the experiments show outstanding performance of RAPIDDOCK, which offers excellent accuracy at unmatched speeds. The backbone transformer architecture is efficient and allows for performing the docking in a single pass. The model is also well-suited for applications to downstream tasks such as predicting the binding strength, e.g., by direct fine-tuning.

**Unlocking Proteome-wide docking** To validate RAPIDDOCK's potential for proteome-wide docking studies, we used our model to run inference for twelve well-studied drugs and toxins, and entire human proteome consisting of 19222 proteins from The Human Protein Atlas (https://proteinatlas.org/) (Uhlén et al., 2015). The whole process took on average 74 seconds per ligand on a machine with eight A100 GPUs.

### 4.2 RESULTS ON APOSTRUCTURES

The Posebusters and DockGen sets contain bound protein structures (i.e., the holostructures), which is a potential data leak, because the models might hypothetically use the knowledge of the deformation of the binding pocket to improve their predictions. Therefore, to better evaluate RAPIDDOCK's practical applicability, we tested its performance on a challenging real-world scenario where the model performs inference given only computationally folded apostructures. We generated such apostructures using AlphaFold-2 for all proteins in the test sets. Since our model was trained to predict distances based on both the holostructures and the aposructutres, we hoped it would be relatively unaffected by any potential information leakage present in holostructures.

We used the apostructures to obtain the distance matrices embeddings, so there is no information leakage for predicting the distances in the resulting complex. To check whether these predicted distances are still accurate, we perform evaluation in the same way as before, using the holostructures

Table 1: Comparison of state-of-the-art molecular docking methods. Details of how we obtained the results can be found in A.3

| DockGen Test Set (*) | | Metrics | |
| --- | --- | --- | --- |
| Model | % RMSD < 2Å↑ | Med. RMSD (Å) ↓ | Avg. Runtime (s) ↓ |
| DiffDock-L | 22.6 | 4.30 | 35.4 |
| **RAPIDDOCK** | **44.0** | **2.83** | **0.05** |
| Posebusters Set | | Metrics | |
| Model | % RMSD < 2Å↑ | Med. RMSD (Å) ↓ | Avg. Runtime (s) ↓ |
| SMINA (**) | 21.9 | 9.26 | 30.6 |
| NeuralPLexer | 27.4 | 3.32 | 3.77 |
| DiffDock-L | 48.8 | 2.13 | 25.0 |
| AlphaFold 3 (***) | **76.9** | **0.72** | 352 |
| **RAPIDDOCK** | 52.1 | 1.90 | **0.03** |

[*] The DockGen test set belongs to the training data of both AlphaFold-3 and NeuralPLexer, so it cannot be used as a reliable evaluation benchmark for those models.
[**] The SMINA predictions were performed on 16 CPUs.
[***] Because docking with AlphFold-3 is not available for public use, the results for AlphaFold-3 are as reported in Abramson et al. (2024).

only for reconstructing the molecule's position. We found that this leads to only a slight deterioration of performance metrics. The percentage of examples with RMSD < 2Å is 51.7% on the Posebusters set, and 42.3% on the DockGen test set, a decrease of only 0.4% and 1.7%, respectively, compared to using holostructures in the input.

### 4.3 ABLATION STUDIES

We performed ablation studies to assess the relative importance of RAPIDDOCK's key features as shown in Table 2. We found protein language model representations (ESM-2 embeddings) to have the strongest impact, followed by protein folding. We speculate that ESM-2 embeddings provide deep contextual understanding of amino-acid sequences due to the model's large scale training. Protein folding, on the other hand, requires the model to internalize the physico-chemical principles governing that process, which evidently aids in modeling also the protein-ligand interactions. Finally, the scaling part of our attention mechanism is also helpful.

Table 2: Ablation study results illustrating the effects of different components on performance in the Posebusters benchmark.

| **Ablation** | % RMSD < 2Å (↑) | Median RMSD (Å) (↓) |
| --- | --- | --- |
| RAPIDDOCK | 52.1 | 1.90 |
| RAPIDDOCK w/o ESM-2 Embeddings | 42.6 | 3.37 |
| RAPIDDOCK w/o Folding Pretraining | 46.4 | 2.61 |
| RAPIDDOCK w/o Attention Scalers | 48.6 | 2.18 |

## 5 RELATED WORK

Traditionally, blind docking consisted of two stages: pocket finding and molecule pose search. Pocket-finding involves identifying cavities in the protein (Le Guilloux et al., 2009; Krivák & Hoksza, 2018; Yan et al., 2022) while molecule pose search aims to find the optimal molecule conformation in the pocket based on scoring functions (Koes et al., 2013; McNutt et al., 2021). While still useful, traditional docking approaches are being gradually replaced by deep learning tools (Corso et al., 2024; Zhou et al., 2024; Clyde et al., 2023).

First successful deep learning models for blind molecular docking applied geometric graph neural networks (Stärk et al., 2022; Lu et al., 2022; Zhang et al., 2022). However, their RMSD-minimizing objective has been considered inadequate for modeling the molecule's position. Instead, DiffDock (Corso et al., 2022) first formulated molecular docking as generative modeling of the ligand pose distribution, with subsequent approaches adapting that paradigm (Lu et al., 2024; Nakata et al., 2023; Corso et al., 2024). However, inference involves costly diffusions, making these approaches two-to-three orders of magnitude slower than non-generative methods such as RAPIDDOCK.

Recently, a different approach to docking has emerged with RoseTTAFold All-Atom (Krishna et al., 2024) and AlphaFold-3 (Abramson et al., 2024). These tools treat the protein-molecule complex as a single structure to be reconstructed from the protein's amino acid sequence and the molecule. AlphaFold-3 claims state-of-the-art results in terms of accuracy (76.9% of examples with RMSD $< 2\text{Å}$ on PoseBusters). However, the docking-while-folding framework is extremely computationally expensive. Consequently, AlphaFold-3's reported inference runtimes are about four orders of magnitude longer than those of RAPIDDOCK run on the same hardware. Although incrementally faster docking-while-folding approaches have been published (Qiao et al., 2024), their speed is still orders of magnitude too slow for allowing large-scale studies envisioned in this paper.

To the best of our knowledge, RAPIDDOCK is the first end-to-end transformer-based model for blind molecular docking. There are, however, notable recent efforts utilizing transformers or transformer-like modules in docking tools. FeatureDock (Xue et al., 2024) is a transformer-based model for pocket-based docking based on a grid of features around the pocket. ETDock (Yi et al., 2023) is composed of several custom modules utilizing the attention mechanism. Our approach also shares several ideas with other existing methods. The equivariance of 3D structures representations is a common feature (Stärk et al., 2022; Lu et al., 2022; Pei et al., 2024). Obtaining the molecule position based on predicted distances is done similarly in TANKBind (Lu et al., 2022) or ArtiDock (Voitsitskyi et al., 2024). Finally, (Gao et al., 2024) most recently proposed a symmetry-invariant loss function similar to ours.

## 6 LIMITATIONS AND FUTURE WORK

RAPIDDOCK achieves excellent accuracy and has a significant speed advantage over other methods. There are several aspects, however, that need further work.

First, we plan to develop a confidence score for RAPIDDOCK's predictions which is important for decision making. While this can be done by training another external light model, ideally we would like to extend RAPIDDOCK to so it can return its own confidence score.

Second, we plan to fine-tune RAPIDDOCK on ligand-protein binding strength prediction, including to identify non-binding ligands. RAPIDDOCK's speed would then allow for efficient identification of all potentially toxic interactions of a drug across the human proteome ($\sim$20k proteins), which is not computationally feasible today and would greatly accelerate the drug discovery process.

Third, we plan to perform more detailed studies of the scaling properties and train larger models on bigger datasets. Among others, we plan to extend pre-training to over 200M protein structures predicted by AlphaFold 2 Varadi et al. (2024), and use the PLINDER (Durairaj et al., 2024) protein-ligand datasets for the main training. This will allow us to extend RAPIDDOCK to tasks such as protein folding or docking without requiring any input protein structure.

## 7 CONCLUSIONS

RAPIDDOCK is a truly *fast* and *accurate* molecular docking model that opens new possibilities for *in silico* drug design. Trained on both protein folding and molecular docking, the model achieves accurate results in molecular docking both on holo- and apostructures.

RAPIDDOCK can already be used for target fishing, that is, screening existing drugs for potential novel targets. We hope that, in the near future, RAPIDDOCK will provide drug designers with a comprehensive view of potential drug interactions across the human proteome. This will unlock important research avenues in biology and machine learning.

Finally, the transformer-based architecture makes the model well-suited for extending pre-training to related biological tasks or fine-tuning on downstream tasks such as predicting toxicity or drug interactions at the cell level, which is one of our current research topics.

ACKNOWLEDGMENTS

To be included in the final version.

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

## A APPENDIX

### A.1 PREPROCESSING DETAILS

**Matching ligand's 3D structure with CCD reference** The PDB files contain 3D structures of proteins and molecules. However, only the 3D position and atomic number of a molecule atoms

are stored explicitly, while the bonds are not directly available. We found that methods of inferring molecule graph from a PDB file available in open sources packages lead to incorrect graphs and SMILES (Weininger, 1988). The correct SMILES is available as CCD code which is also stored in PDB files. A match of the SMILES with the ligand 3D structure from PDB files is performed using Maximum Common Substructure matching (`rdFMCS.FindMCS` from RDKit package (Bento et al., 2020)).

## A.2 ROOT MEAN SQUARE DEVIATION (RMSD) METRIC

The Root Mean Square Deviation (RMSD) is a standard metric for measuring the average distance between the corresponding atoms in two sets of atomic coordinates. For heavy atoms of the ligand, the RMSD is calculated as:

$$\text{RMSD} = \sqrt{\frac{1}{N} \sum_{i=1}^{N} (\mathbf{x}_i - \hat{\mathbf{x}}_i)^2}, \tag{3}$$

where $N$ is the number of heavy atoms, $\hat{\mathbf{x}}_i$ is the position vector of the $i$-th heavy atom in the predicted ligand, and $\mathbf{x}_i$ is the position vector of the corresponding atom in the ground-truth ligand.

To account for permutation symmetries in the ligand, we use the symmetry-corrected RMSD as described in previous studies (Corso et al., 2024).

## A.3 MODEL COMPARISON DETAILS

To obtain reliably comparable inference times for the models in Table 1, we ran DiffDock-L and NeuralPLexer models on our hardware. For this, we used inference scripts available at, respectively, `https://github.com/gcorso/DiffDock` and `https://github.com/zrqiao/NeuralPLexer/tree/main`. The reported metrics come from these evaluation runs. For DiffDock-L, we report metrics on the generated pose with the highest confidence (top-1).

In case of AlphaFold-3, we used the average runtimes reported in Abramson et al. (2024) for the sequence length that was closest to the average sequence length in Posebusters. Because those runtimes were obtained on sixteen A100 GPUs, for a fair comparison, we multiplied them by 16.

In case of SMINA, we ran the blind docking protocole on 16 CPUs with an autobox based on the whole protein structure with a 4Å padding. The program failed for four complexes which were excluded from the analysis. The average runtimes were measured with the `-num_modes=10` option and the RMSD metrics were calculated based on the top-1 conformation.

In reporting runtimes of all methods, we follow the authors of our comparison baseline models and do not include pre-processing. Pre-processing will be negligible in docking studies with thousands of proteins.

## A.4 TRAINING DETAILS

The optimal model checkpoints were selected based on the percentage of examples with RMSD < 2Å on the DockGen validation set. The details of the training parameters are shown in Table 3.

## A.5 MOLECULE POSE RECONSTRUCTION DETAILS

For solving the optimization problem described in Section 2.4, we use the implementation of the L-BFGS algorithm from PyTorch (`torch.optim.LBFGS`), so the optimization is also run on the GPU. We used the following parameters in that function: `tolerance_grad` = 1e−3, `tolerance_change` = 1e−3, `lr` = 1, `max_iter` = 100, `max_eval` = 500, `history_size` = 5, `line_search_fn` = `strong_wolfe`. In particular, using the tolerances of 1e−3 resulted in the same RMSD results on the test sets as with smaller tolerances, but shorter runtimes, on average about two times longer than the transformer forward pass. The optimizer was always able to converge, typically within ten to twenty iterations. We note that although

Table 3: Training Parameters for Pretraining and Fine-Tuning

| Parameter | Folding Pretraining | Docking Training |
|---|---|---|
| Dataset Size | $440k$ | $30k$ |
| Masking Factor | $97\%$ | - |
| Task | Amino acid distance prediction | Molecular docking |
| Epochs | 100 | 200 |
| Batch Size | 8 | 4 |
| Gradient Accumulation Steps | 8 | 16 |
| Optimizer | AdamW | AdamW |
| Learning Rate | $1 \times 10^{-4}$ | $1 \times 10^{-4}$ |
| Max Sequence Length | 1024 | 2760 |
| Weight Decay | 0.01 | 0.01 |
| Precision | bf16 | bf16 |
| Training Duration | 48 hours | 16 hours |

the optimization objective is not smooth, this did not seem to affect the optimizer as we obtained essentially the same results with L1 loss and smooth L1 loss (`nn.L1Loss` and `nn.SmoothL1Loss` in PyTorch).

### A.6 OPTIONAL POST-PROCESSING OF THE RECONSTRUCTED LIGAND

The positions of atoms obtained by minimizing the objective in Equation (2) may not correspond to a chemically valid molecule conformation. We found that the following post-processing step can be employed if such a conformation is desired. The step involves solving an additional optimization problem with an objective function composed of three terms

$$\mathcal{L}^{\text{post}} = \mathcal{L}^r + \mathcal{L}^s + \mathcal{L}^b + \mathcal{L}^h,$$

where

- $\mathcal{L}^r$ is the same as in Equation (2).
- $\mathcal{L}^s$ is defined as the L1 loss between the non-negative entries in the molecule rigid distance matrix described in Section 2.1 and the corresponding predicted distances.
- $\mathcal{L}^b$ is the bond-angle loss defined as the mean value of $1 - \cos(\Delta\alpha)$ where $\Delta\alpha$ is the difference between the predicted and expected bond angle, with expected angles defined as those that are constant across the RDKit conformations. More precisely, the angles are computed based on the molecule rigid distance matrix where only the non-negative entries corresponding to pairs of atoms sharing a bond are considered.
- $\mathcal{L}^h$ is defined in an analogous way for dihedral angles.

The initial guess for the post-processing step is the output of the ligand reconstruction described in Section 2.4. We found that after post-processing, $85\%$ of predictions with RMSD $< 2\text{Å}$ on the Posebusters benchmark are also physically valid as defined in Buttenschoen et al. (2024).

### A.7 EXAMPLES OF GENERATED POSES

Below we present visualizations of selected examples where RAPIDDOCK correctly finds the molecule's pose while AlphaFold 3 fails to do so, or or vice versa.

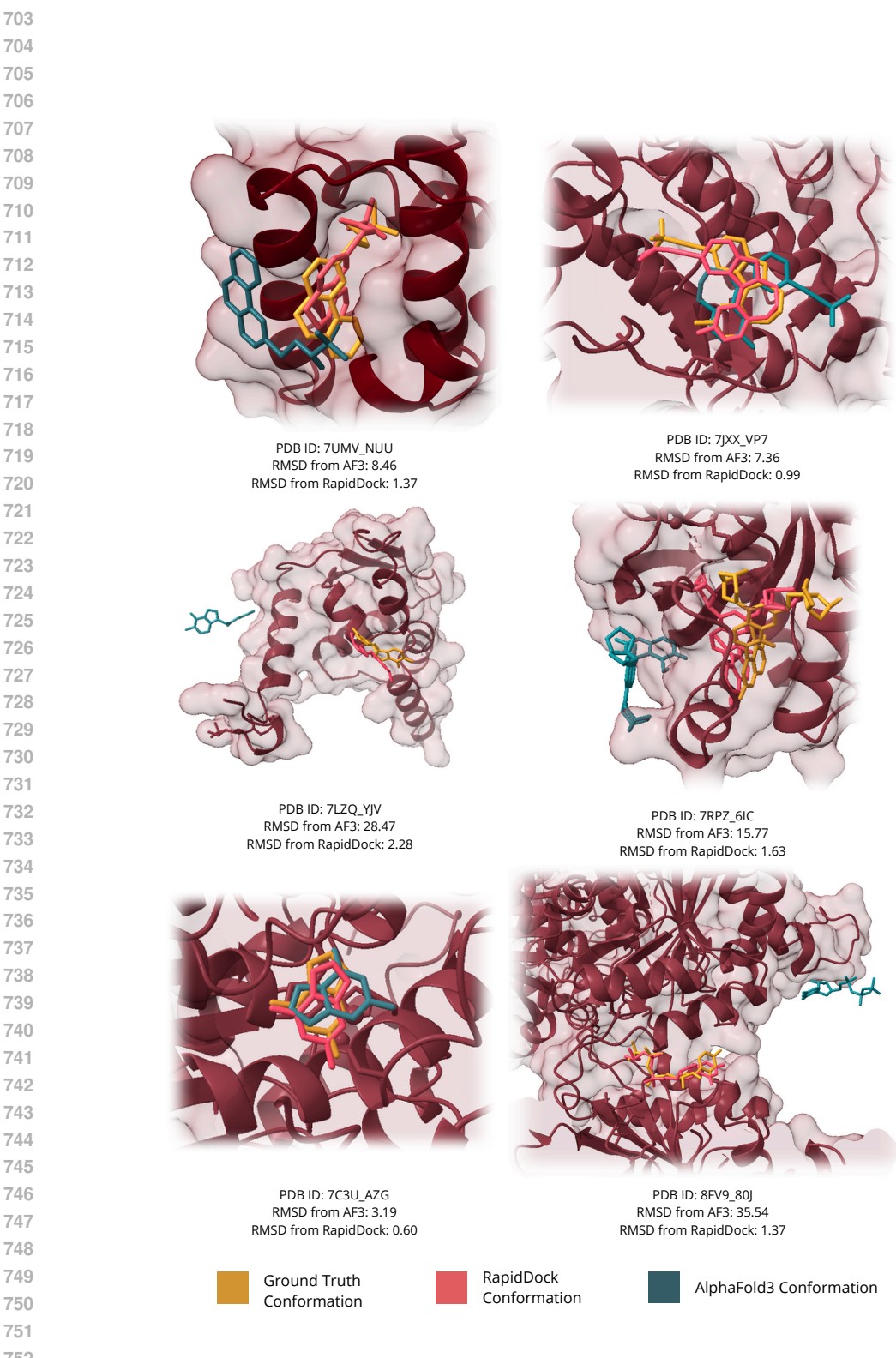

PDB ID: 7UMV_NUU
RMSD from AF3: 8.46
RMSD from RapidDock: 1.37

PDB ID: 7JXX_VP7
RMSD from AF3: 7.36
RMSD from RapidDock: 0.99

PDB ID: 7LZQ_YJV
RMSD from AF3: 28.47
RMSD from RapidDock: 2.28

PDB ID: 7RPZ_6IC
RMSD from AF3: 15.77
RMSD from RapidDock: 1.63

PDB ID: 7C3U_AZG
RMSD from AF3: 3.19
RMSD from RapidDock: 0.60

PDB ID: 8FV9_80J
RMSD from AF3: 35.54
RMSD from RapidDock: 1.37

Ground Truth Conformation    RapidDock Conformation    AlphaFold3 Conformation

PDB ID: 6XBO_5MC
RMSD from AF3: 0.21
RMSD from RapidDock: 1.04

PDB ID: 7SIU_9ID
RMSD from AF3: 0.90
RMSD from RapidDock: 3.11

PDB ID: 8C7Y_TXV
RMSD from AF3: 0.27
RMSD from RapidDock: 0.99

PDB ID: 7BA0_T5H
RMSD from AF3: 0.82
RMSD from RapidDock: 1.98

PDB ID: 8AIJ_M9I
RMSD from AF3: 1.33
RMSD from RapidDock: 3.44

PDB ID: 7KFO_IAC
RMSD from AF3: 1.19
RMSD from RapidDock: 4.27

Ground Truth Conformation    RapidDock Conformation    AlphaFold3 Conformation

