# OpenReview forum: "RapidDock: Unlocking Proteome-scale Molecular Docking"
_ICLR.cc/2025/Conference — Submitted to ICLR 2025_

### Official Review · Reviewer_S2uY · 2024-10-29

**Soundness:** 3
**Presentation:** 3
**Contribution:** 2
**Rating:** 3
**Confidence:** 3

**Summary:**

The paper introduces RapidDock, a fast and accurate transformer-based model for the
blind-docking task. The model predicts interatomic distances. Afterwards, the docked
pose is reconstructed with the L-BFGS algorithm. The authors report a 100x speed-up while
simultaneously improving over commonly used deep learning-based docking models in the
PoseBusters and DockGen datasets. The authors make first experiments at human
proteome-scale docking.

**Strengths:**

RapidDock is fast and accurate, allowing it to tackle human proteome-scale docking studies. The reported speed could enable its use as an oracle function for other DD-related tasks in future work. Ablations on PoseBusters and Dockgen show promising results. The
paper is well written and gives a lot of insights into the modelling and training

**Weaknesses:**

While the method shows good results on a task relevant to computational biology, there is not enough novelty on the ML side that justifies acceptance as a main track contribution. The authors use a standard transformer embedding the ligand and proteins with ESM2 and
employ cross-attention to the distance matrix. I encourage submission to a domain-specific journal.

**Questions:**

Wording:
L. 30 “ … will revolutionize medicine” is an unsupported statement, please reformulate. It would be better to focus on the obstacles that need to be overcome and how you tackle those (like in
your paragraph 2, L. 34 X.).
In l.113f. the authors claim equivariance, however they work with interatomic distances only, making your model invariant.
Permutation loss citation is wrong (e.g. l. 228), cite original paper (Zhu et al. 2022 Direct molecular conformation generation)
In my opinion, l.419f. “the model demonstrates a strong understanding of the physicochemical principles” is a big stretch - how is this supported in your work? Accurate and fast prediction of ligand poses in proteins doesn’t mean that the model understands the physicochemical principles that lead to those poses, and it is debatable whether this is even needed.
Open questions:
Since the prediction time is critical to the paper's claims, I would like to see how much time you spend at inference on average in a) pre-processing, b) prediction, c) reconstruction and d) any form of post-processing.

In l.162f., what is the reasoning for choosing 257 buckets? Are there ablations on higher/lower resolution? How does this affect the performance-inference time tradeoffs? Same questions for the charge embeddings.

---

> ### Author Response · Authors · 2024-11-26
> **Reply to Reviewer S2uY**
>
> Thank you for your comments. Indeed, we would like to use RapidDock as a kind of oracle function in our future work (or to produce embeddings for other models). We are glad you appreciated our efforts to present the ideas in a well-structured way.
>
> To clarify the role of the distance matrices, they are embedded into the self-attention matrices of the encoder (there is no cross-attention). We respectfully disagree with the statement that the embedding of the protein is standard. In fact, this is the key part that enables using the encoder as an end-to-end docking tool for the first time. ESM is an additional external embedding but the precise geometric embedding into the attention matrices is novel. A similar idea has been recently proposed after our submission in Dockformer [1] which is a model for pocket-based docking.
>
> ---
>
> ### Questions:
>
> *Wording: L. 30 “ … will revolutionize medicine” is an unsupported statement, please reformulate. It would be better to focus on the obstacles that need to be overcome and how you tackle those (like in your paragraph 2, L. 34 X.).*
>
> We changed “will” to “could”. The next sentence strongly supports the claim.
>
> ---
>
> *In l.113f. the authors claim equivariance, however they work with interatomic distances only, making your model invariant.*
>
> It’s a fair point that the model is (strictly speaking) invariant (though equivariance is often used interchangeably in the field). We changed the wording.
>
> ---
>
> *Permutation loss citation is wrong (e.g. l. 228), cite original paper (Zhu et al. 2022 Direct molecular conformation generation).*
>
> Thank you for this remark, we fixed the citation to point to the original paper.
>
> ---
>
> *In my opinion, l.419f. “the model demonstrates a strong understanding of the physicochemical principles” is a big stretch - how is this supported in your work? Accurate and fast prediction of ligand poses in proteins doesn’t mean that the model understands the physicochemical principles that lead to those poses, and it is debatable whether this is even needed.*
>
> We toned down the wording.
>
> ---
>
> *Open questions: Since the prediction time is critical to the paper's claims, I would like to see how much time you spend at inference on average in a) pre-processing, b) prediction, c) reconstruction and d) any form of post-processing.*
>
> Following the authors of all our comparison benchmark models, we do not include pre-processing in runtimes. It is dominated by generating conformations which is a highly parallelizable process and will be negligible for docking to thousands of proteins. We do not include pre-processing times also because comparisons to other methods would be problematic. For example, the reported runtimes of AlphaFold 3 include only the GPU wallclock time. NeuralPLexer reports GPU runtimes only as well.
> Reconstruction is included in runtimes and is on average about two times longer than the transformer forward pass as we mention in A.3.
> Optional post-processing is now described in the Appendix A.6. We hope, however, that the model can be directly fine-tuned on downstream tasks or will be used to produce docking embeddings for other models, without forming the poses.
>
> ---
>
> *In l.162f., what is the reasoning for choosing 257 buckets? Are there ablations on higher/lower resolution? How does this affect the performance-inference time tradeoffs? Same questions for the charge embeddings.*
>
> Thank you for this remark. Indeed, these ablations would be desirable. There is no reason inference times would change but performance might indeed be affected. The number 257 was chosen for several reasons:
>
> - An uneven number was chosen because we added one special embedding for unknown distances.
> - The resolution must be sufficiently high to capture nuances in the molecule representation, e.g., to distinguish the length of the double carbon bond from single carbon bond (1.5 vs 1.3).
> - The maximum distance must be sufficient to capture the far-away dependencies interactions between amino acids.
>
> We added some clarifications of these in the paper.
>
> ---
>
> ### References:
>
> [1] Yang, Zhangfan, Junkai Ji, Shan He, Jianqiang Li, Ruibin Bai, Zexuan Zhu, and Yew Soon Ong. "Dockformer: A transformer-based molecular docking paradigm for large-scale virtual screening." *arXiv preprint arXiv:2411.06740* (2024).

---

### Official Review · Reviewer_fj9R · 2024-11-02

**Soundness:** 3
**Presentation:** 2
**Contribution:** 2
**Rating:** 5
**Confidence:** 5

**Summary:**

In this paper, the authors introduce RapidDock, a new approach to molecular docking that leverages a Transformer model. The method includes both ligand atom embeddings and ligand charge embeddings. Protein representations are generated using embeddings from protein amino acids, the ESM-2 PLM, and calculated distance metrics. Instead of deep learning, RDKit-based methods such as MMFF and EDKG, construct the rigid distance matrix for molecules. Trained on the PDBBind and BindingMOAD datasets, RapidDock outperforms DiffDock and other open-source methods on the PoseBuster benchmark in % of ligands in RMSD < 2 Å metric, delivering at least a 100x increase in inference speed.

**Strengths:**

- The authors demonstrate the need for a faster model by outlining the scalability limitations of previous deep learning models.
- Aligned with their motivation, RapidDock shows the ability to perform conformation sampling for molecular docking in GPU inference runtime in approximately one-hundredth of a second per protein-ligand pair.
- In benchmarking with PoseBuster, RapidDock achieves the best performance among open-source codes (noting that AlphaFold 3 is not open source), particularly in the percentage of ligands achieving RMSD < 2 Å.
- The paper provides a clear explanation of how ligand and protein modalities are utilized and fed into the Transformer model.
- For constructing the ligand distance matrix, the authors use a hybrid approach, incorporating physics-based methods from RDKit, such as MMFF and EDKG.
- Additionally, the inclusion of ligand charge embeddings represents another hybrid approach in the model.
- For protein embeddings, the authors showcase the effectiveness of using not only pre-trained models but also their custom-trained ESM-2 models.
- The Transformer architecture employs a non-autoregressive approach with a full attention mask, introducing a new method for molecular docking by incorporating an attention scaler within the attention mechanism.
- The training hyperparameters are shared in detail through comprehensive tables.

**Originality:** RapidDock introduces a unique approach to molecular docking, particularly in terms of preprocessing compared to other DL-based methods. The detailed steps in the ligand and protein embedding process highlight its originality, which the authors further validate through an ablation study.

**Significance:** From a large-scale proteomic perspective, RapidDock is highly scalable and significantly faster in runtime compared to other DL-based and search-based molecular docking methods.

**Weaknesses:**

- The code is not shared.
- Although the method section claims equivariance, it lacks sufficient explanation on this aspect.
- The rationale for using ligand atom charges is not adequately clarified.
- It is unclear why non-fixed distances in the molecule's rigid distance matrix are assigned a value of -1.
- The annotations for distance bias matrices are insufficiently explained; the annotations appear to be included simply because they work, without detailing why they are effective.
- Similarly, the rationale behind RapidDock’s use of attention and charge embeddings, along with their annotations, is not fully addressed.
- The splitting strategy for the training, validation, and test sets is not sufficiently described.
- The parameter comparison in benchmarking does not seem fair; DiffDock results are compared with 30 million parameters, while RapidDock has 60 million parameters.
- The RMSD metric, commonly used in molecular docking and Structure-Based Drug Design (SBDD), does not always yield bioactively, physically, or chemically plausible structures, as shown by Posebuster[1], PoseCheck[2], PoseBech[3], and CompassDock[4]. Including these metrics in the benchmark would strengthen the study.
- Appendix A.6 lacks a comparison between DiffDock and NeuralPLexer examples.
- The extent of ligand filtering during ligand preparation is not sufficiently discussed.

**Quality:** Although the authors claim this is the first use of a Transformer-based model in blind docking, ETDock[5] and FeatureDock[6] have previously used Transformers for molecular docking; however, these methods are not mentioned in the paper. Additionally, the benchmarking is limited to comparisons with only a few popular methods. In the conclusion, the authors state that:

>"... the model demonstrates a strong understanding of the physicochemical principles behind forming biological structures,"

yet no bioactivity or physicochemical analyses, as discussed earlier, have been conducted to support this claim.

**Clarity:** The paper contains numerous grammatical errors that detract from readability and should be carefully revised. Placing the Related Work section after the Experiments section disrupts the flow, and the method annotations are not clearly explained.

**Reproducibility:** As the code is not shared, it is currently impossible to test whether it performs as reported. If the code were provided, I would be able to review, test, and reassess my evaluation (including scoring and comments) accordingly.

### **References**
[1] Martin Buttenschoen, Garrett M Morris, and Charlotte M Deane.  Posebusters:  Ai-based docking methods fail to generate physically valid poses or generalise to novel sequences. Chemical Science, 15(9):3130–3139, 2024.

[2] Charles Harris, Kieran Didi, Arian R Jamasb, Chaitanya K Joshi, Simon V Mathis, Pietro Lio, and Tom Blundell. Benchmarking generated poses: How rational is structure-based drug design with generative models? arXiv preprint arXiv:2308.07413, 2023.

[3] Alex Morehead, Nabin Giri, Jian Liu, Jianlin Cheng. Deep Learning for Protein-Ligand Docking: Are We There Yet? arXiv preprint arXiv:2405.14108, 2024.

[4] Ahmet Sarigun, Vedran Franke, Bora Uyar, Altuna Akalin. CompassDock: Comprehensive Accurate Assessment Approach for Deep Learning-Based Molecular Docking in Inference and Fine-Tuning. arXiv:2406.06841, 2024.

[5] Yiqiang Yi, Xu Wan, Yatao Bian, Le Ou-Yang, Peilin Zhao. ETDock: A Novel Equivariant Transformer for Protein-Ligand Docking. arXiv:2310.08061, 2023.

[6] Mingyi Xue, Bojun Liu, Siqin Cao, Xuhui Huang. FeatureDock: Protein-Ligand Docking Guided by Physicochemical Feature-Based Local Environment Learning using Transformer. ChemRxiv

**Questions:**

- When comparing runtime in inference mode, did you preprocess the protein, or was the comparison done solely for molecule conformation?
- Did you use the same time-splitting approach as previous methods, such as DiffDock and NeuralPlexer, for the dataset?
- In the statement,
> "Only the fixed distances across the molecule’s possible conformations are recorded, and others are denoted by a special value of −1,"

  Could you clarify why you selected -1 as the special value?
- In the RapidDock attention section, you state,
> "First, we multiply the attention scores corresponding to input pairs with known distances (i.e., ligand-ligand within a rigid part and protein-protein) by a learnable scalar $s_m$, one for each layer $m$."

  Did you also apply this to protein-ligand pairs? If not, could you explain why?
- For inference, do you generate one conformation per runtime for each protein-ligand pair?

---

> ### Author Response · Authors · 2024-11-26
> **Reply to Reviewer fj9R**
>
> Thank you for carefully reading our manuscript and listing multiple strengths of our method. We also believe that our embeddings of the protein and ligand 3D structures are the key to success.
> We appreciate your pointing out the need for improving some of the descriptions of technical details in the paper. We tried to improve them.
>
> Regarding the differences in terms of the number of parameters, we do not think it is particularly relevant here because in practice only speed and accuracy matter. Our main claim is that a properly designed transformer can perform competitively while being much faster than models based on generative modeling.
>
> Your comment about the RMSD metric being imperfect is of course correct. The actual molecule poses can be obtained via post-processing, which we now describe in Appendix A.6. However, we hope that RapidDock will be fine-tuned on downstream tasks directly, or used for obtaining docking embeddings, without forming any poses explicitly.
>
> In Appendix A.7, we now also include examples where RapidDock performs worse than AlphaFold 3. We think a comparison to the most accurate method is sufficient.
>
> Thank you for referring to ETDock and FeatureDock. While these methods use transformers in their pipeline (as do the models in our baseline comparisons), we do not think they are end-to-end blind docking tools fully based on a single transformer, although we admit that such a distinction is not clear-cut. ETDock is composed of several custom modules (Feature Processing, TAMformer, Attention Layer, Message Layer, etc). FeatureDock is not a blind docking tool because the grid around the pocket is input to the model. We agree that these methods should be cited though, and we added a mention of these methods in the paper.
>
> Finally, we are confident that the paper does not contain major grammatical errors. We do acknowledge, though, that there may be occasional typos or misplaced punctuation. We placed the Related Work section after Experiments to better describe other methods as they relate to ours.
>
> ---
>
> ### Questions:
>
> *When comparing runtime in inference mode, did you preprocess the protein, or was the comparison done solely for molecule conformation?*
>
> The protein requires minimal preprocessing (only computing the distances), though ESM embeddings need to be ready. The number of proteins in the human body is limited to only about 20,000, however, and the preprocessing can be done “once and for all,” so it can be neglected.
>
> ---
>
> *Did you use the same time-splitting approach as previous methods, such as DiffDock and NeuralPlexer, for the dataset?*
>
> No, because that approach is known to be very leak-prone. See, for example, [1].
>
> ---
>
> *In the statement,
> "Only the fixed distances across the molecule’s possible conformations are recorded, and others are denoted by a special value of −1,"
> could you clarify why you selected -1 as the special value?*
>
> We wanted to use a value that does not correspond to any distance. This value indicates that:
> 1. The part of the first term inside softmax corresponding to those distances should not be modified (see the definition of $S_m$), and
> 2. The distance is larger than the maximum distance of 16 (see the definition of $b(x)$).
>
> The special value could be any negative number. We added a clarification.
>
> ---
>
> *In the RapidDock attention section, you state,
> "First, we multiply the attention scores corresponding to input pairs with known distances (i.e., ligand-ligand within a rigid part and protein-protein) by a learnable scalar $s_m$, one for each layer $m$."
> Did you also apply this to protein-ligand pairs? If not, could you explain why?*
>
> The scalar $s_m$, together with $z_m$, controls how much the final score is affected by the original score and how much by the distance bias. The protein-ligand distances are unknown, so the model should rely solely on the original attention score. That part could be multiplied by another scalar, but the overall effect would be the same because we already have two scalars, $z_m$ and $s_m$. We added a clarification.
>
> ---
>
> *For inference, do you generate one conformation per runtime for each protein-ligand pair?*
>
> All conformations need to be generated at inference.
>
> ---
>
> ### References:
>
> [1] Li, Jie, Xingyi Guan, Oufan Zhang, Kunyang Sun, Yingze Wang, Dorian Bagni, and Teresa Head-Gordon. "Leak proof PDBBind: A reorganized dataset of protein-ligand complexes for more generalizable binding affinity prediction." *arXiv preprint arXiv:2308.09639* (2023).

---

### Official Review · Reviewer_1PMn · 2024-11-03

**Soundness:** 1
**Presentation:** 1
**Contribution:** 3
**Rating:** 3
**Confidence:** 3

**Summary:**

This work proposed a fast and reliable prediction method, RapidDock, for protein-ligand binding poses. Most previous methods employed in this work as base-lines used diffusion models, leading to large computational costs. However, RapidDock is based on a transformer and so much faster than the base-line models. The benchmark studies on PoseBusters and DockGen show that RapidDock is not only fast but also far more accurate than others except AlphaFold3. While the performance of the proposed method seems competitive, there are several issues that need to be addressed.

**Strengths:**

1. This work shows the possibility of transformer-based approaches for binding structure predictions, while most previous works are based on diffusion methods.
2. The proposed method outperformed the popular base-line, DiffDock-L, in the two benchmark studies, while its computational time for predictions is much faster than those of all base-line models.

**Weaknesses:**

1. The proposed method needs to generate 96 molecular conformations for each molecule and analyze the conformers to obtain its distance matrix.
2. The experiment in Section 4.2 seems meaningless because it uses holostructures when it predicts binding poses.
3. The title and introduction parts emphasize the importance of proteome-wide docking, but this work does not provide any meaning results regarding that.
4. Technical details of the proposed method are insufficient.

**Questions:**

1. DiffDock-L (also AlphaFold3 and NeuralPLexer) is a generative approach, so it gives multiple binding poses according to their probabilities. Therefore, its prediction accuracy can be improved by including multiple different poses (Top-n poses) in the evaluation. However, RapidDock is deterministic, so it gives only a single output. It is understood that the authors want to emphasize computational speeds, but it seems they also need to discuss the accuracy aspects for a fair comparison.
2. Since RapidDock requires conformation searches for each molecule before docking, the authors need to clarify whether the reported computational times include the time for conformer searches or not.
3. The proposed method has been compared with AlphaFold3 and NeuralPLexer. However, the former performs a rigid docking, whereas the latter predicts binding poses only from protein sequences and molecular graphs. Therefore, the prediction complexity of RapidDock is much lower than that of the baseline models, so the direct comparison between them is less meaningful.  The authors need to clarify this fact in the introduction or result sections. The current form may cause undesirable confusion to potential readers.
4. In the reconstruction of ligand location, predicted distances between ligands and also between ligand-protein may not precisely match with the coordinates of a single pose. If this is the case, the authors should elaborate more details about this process. Does it require a kind of post-processing?
5. The authors greatly emphasize the importance of proteome-wide docking, but this work does not provide any meaningful analysis except the computational time, which can be readily estimated without performing the actual calculations. The authors may add an additional study verifying that the proposed method can indeed provide meaningful results from the proteome-wide docking. Otherwise, they need to tone down their argument from the title and introduction parts.
6. Appendix A.6 shows the examples of 3D structures predicted by RapidDock and AlphaFold3. The authors deliberately selected specific examples where RapidDock outperformed AlphaFold3, while they admit that the latter is far better than the former on average. AlphaFold3 even predicted those structures from the sequence-level information. These examples may lead to misunderstanding that RapidDock works better than AlphaFold3. The authors need to provide examples where RapidDock fails while AlphaFold3 succeeds.

---

> ### Author Response · Authors · 2024-11-26
> **Reply to Reviewer 1PMn**
>
> Thank you for carefully reading our manuscript and providing constructive feedback. We acknowledge the need for generating conformations. However, this is embarrassingly parallel and will be negligible in docking studies with thousands of proteins. None of the baseline models we compare to report pre-processing times.
>
> We are aware that the experiment in Section 4.2 only confirms the quality of the predicted protein-ligand distances. A better experiment would align the entire predicted structure with the bound complex and compute RMSD or LDDT-PLI metrics only then. This is part of our current work.
>
> ---
>
> ### Questions:
>
> *DiffDock-L (also AlphaFold3 and NeuralPLexer) is a generative approach, so it gives multiple binding poses according to their probabilities. Therefore, its prediction accuracy can be improved by including multiple different poses (Top-n poses) in the evaluation. However, RapidDock is deterministic, so it gives only a single output. (...).*
>
> We do not believe that the non-deterministic nature is a shortcoming of our method, especially for high-throughput studies, though various approaches to generating a collection of poses could be considered. The results for other methods were obtained with the parameters recommended by their authors. In particular, DiffDock generates ten poses and chooses the most likely pose via a ranking module.
>
> ---
>
> *Since RapidDock requires conformation searches for each molecule before docking, the authors need to clarify whether the reported computational times include the time for conformer searches (...).*
>
> As mentioned above, the runtimes do not include preprocessing. For studies with thousands of proteins, they should be negligible. Moreover, the comparisons to other methods would be problematic. We followed the authors of our comparison baselines and do not report preprocessing times. For example, the reported runtimes of AlphaFold 3 include only the GPU wallclock time. NeuralPLexer or DiffDock authors also report GPU inference runtimes only. We added a clarification of what runtimes are reported in Appendix A.3.
>
> ---
>
> *The proposed method has been compared with AlphaFold3 and NeuralPLexer. However, the former performs a rigid docking, whereas the latter predicts binding poses only from protein sequences and molecular graphs. (...)*
>
> You are absolutely right to point out that AlphaFold3 and NeuralPLexer perform docking-while-folding, which is a more general task. We write about it in detail in the Related Work section. However, it is particularly important to compare RapidDock to the tools that achieve state-of-the-art accuracy whatever they may be.
>
> ---
>
> *In the reconstruction of ligand location, predicted distances between ligands and also between ligand-protein may not precisely match with the coordinates of a single pose. (...) Does it require a kind of post-processing?*
>
> Thank you for pointing this out. This is a very good point. The results are obtained without any post-processing. Such post-processing can be performed, though, which we now describe in Appendix A.6. However, we believe that in practice – being fully based on a transformer – the model is well-suited for fine-tuning without explicitly forming the poses.
>
> ---
>
> *The authors greatly emphasize the importance of proteome-wide docking, but this work does not provide any meaningful analysis except the computational time, (...). The authors may add an additional study verifying that the proposed method can indeed provide meaningful results from the proteome-wide docking (...).*
>
> Demonstrating the importance of proteome-wide docking is a topic of our current research. It is a fact, however, that the number of screened proteins for assessing toxicity of new drugs nowadays is limited because of computational constraints and most drugs fail even the first phases of clinical trials. We believe that to answer why this is the case, one needs a full picture of the given molecule’s effects. We think RapidDock is one important step toward enabling that. This approach is already partially utilized in genomics, where similarity of drug effect is measured by the similarity of their expression profiles on a pre-defined set of genes. Demonstrating actual use cases for proteome-wide docking is part of our current work.
>
> ---
>
> *Appendix A.6 shows the examples of 3D structures predicted by RapidDock and AlphaFold3. The authors deliberately selected specific examples where RapidDock outperformed AlphaFold3, while they admit that the latter is far better than the former on average. AlphaFold3 even predicted those structures from the sequence-level information. (...).*
>
> We include such examples in the new version of the paper.
>
> ---
>
> ### References
>
> [1] Li, Jie, Xingyi Guan, Oufan Zhang, Kunyang Sun, Yingze Wang, Dorian Bagni, and Teresa Head-Gordon. "Leak proof PDBBind: A reorganized dataset of protein-ligand complexes for more generalizable binding affinity prediction." *arXiv preprint arXiv:2308.09639* (2023).

---

### Official Review · Reviewer_KqLH · 2024-11-03

**Soundness:** 3
**Presentation:** 4
**Contribution:** 2
**Rating:** 6
**Confidence:** 4

**Summary:**

The authors tackle the problem of proteome-scale docking, the goal of which is predicting the binding pose of a ligand against many thousands of proteins. To do this, they develop an equivariant Transformer model (RapidDock) for dramatically accelerating docking compared to previous diffusion or GNN-based approaches. The model takes various features from the protein and ligand as input, and outputs a prediction of the binding pose of the ligand. The results show that RapidDock achieves 100x faster runtimes than three competing deep learning methods, while retaining equivalent accuracy (except to AlphaFold 3, which is much slower but has much better accuracy).

**Strengths:**

* Clear presentation of methods and results.
* Novel application of transformer architecture to docking, which results in much faster inference.
* Reasonable design choices in model. These include the addition of features for ligand atom charges and the use of a pre-trained protein language model. The scaling of the attention vector based on distance also seems well-motivated.
* Strong results on two distinct datasets, achieving a better success rate than two competitive deep learning methods at a fraction of the cost. While RapidDock has significantly worse accuracy on Posebusters compared to AlphaFold 3, I do not think this is a negative, because as the authors note AlphaFold 3’s speed is not suitable for large-scale docking. Additionally, my understanding is that AlphaFold-3 performs energy minimization as a post-processing step, which would give it an unfair advantage compared to RapidDock.
* Ablations of various model components help show the benefits of each design choice.

**Weaknesses:**

* Motivation behind the problem setting is unclear. The authors address proteome-scale docking because any protein in the human proteome could be a potential *off-target* (a term of art that should probably be included in the paper) of a drug. Thus, docking against all proteins and then predicting affinity in a downstream task would detect these potential off-targets before they are discovered in later preclinical or clinical testing. However, I am not convinced that docking to each protein in the proteome is necessary to detect off-target effects. Pharmacologists often screen a drug against a limited number of safety-relevant proteins (in the hundreds), such as G-protein coupled receptors or ion channels, that are frequent off-targets [1, 2]. This is usually sufficient to detect many clinical issues beforehand, and it is not obvious that just considering more potential off-targets would further reduce the rate of adverse effects occurring (which for example may be due to more complex issues, such as toxicity of metabolic products or on-target unwanted effects). Could the authors provide a better argument for why it is important to screen a drug against all potential protein targets?
[1] Bendels et al. “Safety screening in early drug discovery: An optimized assay panel.“ J Pharmacol Toxicol Methods 2019.
[2] Peters et al. “Can we discover pharmacological promiscuity early in the drug discovery process?” Drug Discovery Today 2012.

* Limited baseline comparisons. I think the most important baseline to include would be AutoDock Vina (or another similar docking program). Despite not being deep learning-based, Vina is relatively fast and the current state-of-the-art in applied fields. Practitioners conducting large-scale docking would likely use Vina, so including results on this baseline is important. An additional deep-learning based docking model, such as TANKBind, would also improve the strength of the results, but it is not as critical.

* Some of the design choices are not well-explained. For example, it is not clear to me why discretized charge embeddings are used for the ligand atoms instead of simply providing the charge scalar as an input. It is also not clear what role the distance bias matrices play.

**Questions:**

* Is there a better way to determine which atoms are rigid and which are flexible? For example, AutoDock Vina determines if bonds are rotatable using a simple chemical definition, which dictates which atoms are flexible and which are not. Just searching through a lot of generated conformations seems like it might miss bonds that only rotate when exposed to external charges.
* Is this model potentially applicable to *target fishing*? Target fishing is the process of taking an existing drug compound and evaluating it against a large number of potential proteins to see if it can target them. This can be applied for drug repurposing, which is the use of currently approved drugs against new indications based on previously unknown binding against a new protein. This is potentially a strong application of the proposed method, but I do not see it explicitly mentioned in the paper anywhere.

---

> ### Author Response · Authors · 2024-11-26
> **Reply to reviewer KqLH**
>
> We are truly thankful for your insightful comments. We are glad that you appreciated our choice to pre-train the model on protein folding and represent the protein and ligand 3D data as inductive biases in the attention matrices. We also think these are the key features of RapidDock. We are also grateful for mentioning the clarity of the presentation.
>
> Thank you for noting that in modern drug design molecules are screened against a relatively small number of proteins for assessing toxicity. This is precisely our motivation for developing a model that can quickly dock molecules to all proteins. The number of screened proteins nowadays is limited because of computational constraints. We believe this is the reason most drugs fail even the first phases of clinical trials. To confirm it, one needs a full picture of the given molecule’s effects. For example, a ligand may indirectly cause side effects by yet unknown protein cascades. Or, as you have pointed out, the side effects may be caused by the metabolic products of the drug, which again calls for a fast docking method in particular. Apart from toxicity, a quick binding tool can also allow studying potential beneficial effects, such as decreasing inflammation or speeding up proliferations in the metabolome due to binding to specific receptors, etc.
>
> We appreciate your suggesting a comparison to classical methods. We now include a comparison to SMINA in the paper. We note that there are many papers that include comparisons of deep learning methods to classical tools. The reported speeds of the latter methods are, however, consistently slower than those of most deep learning methods.
>
> Regarding the charge embeddings, we believe that including “per token” numerical values via embeddings is a standard approach, though we admit there are other possible choices. The distance matrices are needed to encode the 3D structure of the protein and the ligand within the transformer in a way that is invariant to translations, reflections, or rotations. The distances describe a protein-protein or ligand-ligand pairwise all-to-all property, so the attention matrix is an appropriate place to include them. We added a clarification of that point in the paper.
>
> ---
>
> ### Questions:
>
> *Is there a better way to determine which atoms are rigid and which are flexible? For example, AutoDock Vina determines if bonds are rotatable using a simple chemical definition, which dictates which atoms are flexible and which are not. Just searching through a lot of generated conformations seems like it might miss bonds that only rotate when exposed to external charges.*
>
> Thank you for this question. For sure, the representation is not perfect and could be improved in the future to better capture such aspects. On the other hand, sampling the conformations is implicitly performed with energy weighting. As a result, the distribution of conformations, and therefore the distances obtained, resembles the one found in reality.
>
> ---
>
> *Is this model potentially applicable to target fishing? Target fishing is the process of taking an existing drug compound and evaluating it against a large number of potential proteins to see if it can target them. This can be applied for drug repurposing, which is the use of currently approved drugs against new indications based on previously unknown binding against a new protein. This is potentially a strong application of the proposed method, but I do not see it explicitly mentioned in the paper anywhere.*
>
> Yes, in fact, that is one of the applications the model is well-suited for. We included a mention of target fishing in the paper.

---

> > ### Comment · Reviewer_KqLH · 2024-11-29
> >
> > Thank you for your reply, and for running SMINA as a baseline. These new results show that RapidDock outperforms both deep learning and classical docking methods, and therefore I increase my score from 5 to 6. RapidDock has strong performance given its short runtime, so I think it could be practically quite useful.
> >
> > >  The number of screened proteins nowadays is limited because of computational constraints. We believe this is the reason most drugs fail even the first phases of clinical trials.
> >
> > I'm still not convinced that the reason most drugs fail early clinical trials is because there was an off-target protein that was missed in an earlier stage of development. It would be good if the authors could provide some citation for this claim, or an argument based on data from clinical trials. Otherwise, the motivation behind the problem setting remains unclear to me.

---

### Author Response · Authors · 2024-11-26

# Response to Reviewers

Dear Reviewers,

Thank you for taking the time to read our manuscript and providing invaluable feedback.

We especially appreciate the warm comments regarding the originality of the protein and ligand representations within the transformer. We are also glad that many of you praised the clarity and structure of our manuscript.

We also accept and deeply value your constructive critical remarks. For sure, there are several aspects of the method that need further effort before the proteome-based docking can be fully unlocked. Several points that you raise are topics of our current work. We tried to address as many of them as possible at this stage. The changes to the manuscript are summarized as follows:

## List of Changes

- We changed the wording in the first sentence.
- We fixed the results obtained for DiffDock-L on Neuralplexer on the Posebusters sets – there was an error due to a bug in our processing of that dataset for those models. DiffDock-L’s results on Posebusters now align better with those reported in their paper. We slightly changed the wording accordingly.
- We changed “equivariant” to “invariant” and improved the explanation of why the model is invariant.
- We described the motivation for embeddings of atom charges.
- We added explanations on the choice of the special value in the distance bias matrix.
- We elaborated on the choice of the number of buckets and maximum value in the $b$ function.
- We expanded on our motivation for including the distance information within the attention matrices.
- We added an explanation of the role of the scalers in our attention mechanism.
- We fixed the reference to the paper that introduced Permutation Loss.
- We added a comparison to a classical docking tool, SMINA.
- In *Related Work*, we included references to FeatureDock and ETDock.
- We changed wording in *Conclusions*.
- We added a mention of target fishing in *Conclusions*.
- We provided more details about how runtimes were reported in Appendix A.3.
- We added a section in the Appendix on the optional post-processing of the ligand.
- We extended the illustration of the docked molecules to include examples where AlphaFold 3 outperforms RapidDock.

Thank you once again for your detailed review and suggestions, which have greatly contributed to the improvement of our manuscript.

---

### Meta-Review · Area_Chair_QDbC · 2024-12-20

**Metareview:**

The paper proposes a rapid docking method based on a Transformer. The model takes as input a ligand and a protein, both of which are tokenized, and their 3D structures are passed through all self-attention layers. The protein is embedded using ESM-2.

The method achieves strong empirical results, and the reviewers appreciated this aspect. Overall, the work is very well executed. The training choices, such as the use of AlphaFold2-generated structures or biasing self-attention with distance, are generally intuitive. (Additional ablations would be very useful.)

However, three out of four reviewers voted for rejection.

A significant drawback of the manuscript is its limited novelty. The idea of using deep learning models to directly predict poses and/or docking scores has been explored before, as noted by one of the reviewers. UniMol2 is one such comparison point.

Another major critique focused on the limited evaluation. More detailed analysis of poses or comparisons to more docking baselines were suggested. One reviewer specifically proposed using a time-based split. It would also be useful to consider other routes to accelerating existing docking methods (e.g., distilling DiffDock).

These lower-level comments on the evaluation scheme reflect a broader issue: the paper does not clearly demonstrate its intended use. Two reviewers expressed doubts about the possibility of proteome-wide docking. For example, it is not fully clear how broadly the method generalizes to novel ligand structures or different protein families, which might be crucial depending on the use case. The paper would benefit from a clearer explanation of the model’s practical applications, ideally supported by an actual demonstration.

All in all, the paper fell slightly short of the acceptance threshold. While its strong empirical results were acknowledged, it did not fully clarify how the model would be used in practice. At this stage, I am recommending rejection. I hope these comments will be helpful in improving the paper.

**Additional Comments On Reviewer Discussion:**

During the rebuttal, reviewers raised concerns about the limited novelty of the Transformer-based architecture, incomplete benchmarking against classical docking tools like AutoDock Vina, and the scalability claims for proteome-wide docking. The authors addressed these by including additional baselines such as SMINA, and clarifying the choice of ligand embeddings and distance matrices. While these changes improved the presentation and partially addressed weaknesses, key issues regarding concerns about generalization to diverse protein families (stemming from limitations of the evaluation), novelty and practical validation of proteome-wide docking remained unresolved.

---

### Decision · Program_Chairs · 2025-01-22

Reject